# One is not enough: On the effects of reference genome for the mapping and subsequent analyses of short-reads

**Carlos Valiente-Mullor**[1], **Beatriz Beamud**[1]*, **Iván Ansari**[1], **Carlos Francés-Cuesta**[1], **Neris García-González**[1], **Lorena Mejía**[1,2], **Paula Ruiz-Hueso**[1], **Fernando González-Candelas**[1,3]*

**1** Joint Research Unit "Infection and Public Health" FISABIO-University of Valencia, Institute for Integrative Systems Biology (I2SysBio), Valencia, Spain, **2** Instituto de Microbiología, Colegio de Ciencias Biológicas y Ambientales, Universidad San Francisco de Quito, Quito, Ecuador, **3** CIBER in Epidemiology and Public Health, Valencia, Spain

* beatriz.beamud@uv.es (BB); fernando.gonzalez@uv.es (FG-C)

**Data Availability Statement:** The authors confirm that all data underlying the findings are fully available without restriction. All relevant data are

## Abstract

Mapping of high-throughput sequencing (HTS) reads to a single arbitrary reference genome is a frequently used approach in microbial genomics. However, the choice of a reference may represent a source of errors that may affect subsequent analyses such as the detection of single nucleotide polymorphisms (SNPs) and phylogenetic inference. In this work, we evaluated the effect of reference choice on short-read sequence data from five clinically and epidemiologically relevant bacteria (*Klebsiella pneumoniae*, *Legionella pneumophila*, *Neisseria gonorrhoeae*, *Pseudomonas aeruginosa* and *Serratia marcescens*). Publicly available whole-genome assemblies encompassing the genomic diversity of these species were selected as reference sequences, and read alignment statistics, SNP calling, recombination rates, d*N*/d*S* ratios, and phylogenetic trees were evaluated depending on the mapping reference. The choice of different reference genomes proved to have an impact on almost all the parameters considered in the five species. In addition, these biases had potential epidemiological implications such as including/excluding isolates of particular clades and the estimation of genetic distances. These findings suggest that the single reference approach might introduce systematic errors during mapping that affect subsequent analyses, particularly for data sets with isolates from genetically diverse backgrounds. In any case, exploring the effects of different references on the final conclusions is highly recommended.

## Author summary

Mapping consists in the alignment of reads (i.e., DNA fragments) obtained through high-throughput genome sequencing to a previously assembled reference sequence. It is a common practice in genomic studies to use a single reference for mapping, usually the 'reference genome' of a species—a high-quality assembly. However, the selection of an optimal reference is hindered by intrinsic intra-species genetic variability, particularly in bacteria.

within the paper and its Supporting information files. Complete pipeline is available on Github (https://github.com/cvmullor/reference) to be run as a single script, so that the analyses conducted in this work could be easily reproduced on any dataset.

**Funding:** This project was partly funded by projects BFU2017-89594R from MICIN (Spanish Government) and PROMETEO2016-0122 (Generalitat Valenciana, Spain). WGS was performed at Servicio de Secuenciación Masiva y Bioinformática de la Fundación para la Investigación Sanitaria y Biomédica de la Comunitat Valenciana (FISABIO) and co-financed by the European Union through the Operational Program of European Regional Development Fund (ERDF) of Valencia Region (Spain) 2014-2020. CV is recipient of contract FPU2018/02579, BB of contract FPU2016/02139 and CF of FPI contract BES-2015-074204 from MICIN (Spanish Government). LM benefits of a fellowship from Fundación Carolina. The funders had no role in study design, data collection and analysis, decision to publish, or preparation of the manuscript.

**Competing interests:** The authors have declared that no competing interests exist.

It is known that genetic differences between the reference genome and the read sequences may produce incorrect alignments during mapping. Eventually, these errors could lead to misidentification of variants and biased reconstruction of phylogenetic trees (which reflect ancestry between different bacterial lineages). To our knowledge, this is the first work to systematically examine the effect of different references for mapping on the inference of tree topology as well as the impact on recombination and natural selection inferences. Furthermore, the novelty of this work relies on a procedure that guarantees that we are evaluating only the effect of the reference. This effect has proved to be pervasive in the five bacterial species that we have studied and, in some cases, alterations in phylogenetic trees could lead to incorrect epidemiological inferences. Hence, the use of different reference genomes may be prescriptive to assess the potential biases of mapping.

## Introduction

The development and increasing availability of high-throughput sequencing (HTS) technologies, along with bioinformatic tools to process large amounts of genomic data, has facilitated the in depth study of evolutionary and epidemiological dynamics of microorganisms [1–3]. Whole-genome sequencing (WGS)-based approaches are useful to infer phylogenetic relationships between large sets of clinical isolates [4–7], showing improved resolution for molecular epidemiology [8–11] compared to traditional typing methods [12–14]. Short-read mapping against a single reference sequence is a commonly used approach in bacterial genomics for genome reconstruction of sequenced isolates and variant detection [4,6,15–17]. Nevertheless, there are grounds for suspecting that this approach might introduce biases depending on the reference used for mapping. Most of these errors originate in the genetic differences between the reference and the read sequence data [18–21], and they can affect subsequent analyses [22–28]. These include the identification of variants throughout the genome (mainly single nucleotide polymorphisms [SNPs]) and phylogenetic tree construction, which are essential steps for epidemiological and evolutionary inferences.

Sequencing status, completeness, assembly quality and annotation are relevant factors in reference selection, which explain the widespread use of the NCBI-defined reference genome of a species for mapping [26,28]. However, these criteria do not necessarily account for the amount of genetic information shared between the reference and subject sequences [29], neither the intrinsic genomic variability of the different bacterial species, which is reflected in their pangenomes (i.e., the total gene set within a species or within a particular sequence data set) [30]. It has been suggested that the impact of reference selection in clonal bacteria such as *Mycobacterium tuberculosis* [31] could be ameliorated by its limited variability at the intra-species level [25,28], although its effects on epidemiological inferences have been described [32]. In contrast, we expect a greater impact of reference choice in species with open pangenomes (e.g., *Pseudomonas aeruginosa* [33]) and/or highly recombinogenic bacteria (e.g., *Neisseria gonorrhoeae* [34] or *Legionella pneumophila* [35]). In spite of the awareness of the problem of reference selection considering the high genomic diversity of most bacterial species, systematic studies on the effect of reference choice in bacterial data sets are still missing, particularly if we are concerned with the consequences on epidemiological or evolutionary inferences. In addition, previous studies considering reference selection explicitly have been mainly focused on biases in SNP calling [23,24,28] and have not addressed other possible implications.

*De novo* assembly of read sequence data dispenses with the need of using a reference genome. However, this requires higher sequencing coverage and longer reads in order to

obtain enough read overlap at each position of the genome. Therefore, obtaining unfinished or fragmented assemblies is a major drawback, particularly when using short-reads (which still are the most frequently used in HTS-based studies) [36]. Complementarily, *de novo* assembled isolates could be used as reference genomes if previously assembled, high-quality references are found to be suboptimal in terms of genetic relatedness to the newly sequenced isolates [12,32,37]. However, this solution still has to deal with the additional costs of long-read sequencing and mapping errors derived from using a low-quality or fragmented reference.

In this work, we have analyzed the effect of reference selection on the analysis of short-read sequence data sets from five clinically and epidemiologically relevant bacteria (*Klebsiella pneumoniae*, *Legionella pneumophila*, *Neisseria gonorrhoeae*, *Pseudomonas aeruginosa* and *Serratia marcescens*) with different core and pangenome sizes [38–41]. WGS data sets were mapped to different complete and publicly available reference genomes, encompassing the currently sequenced genomic diversity of each species. We have studied the effect of reference choice on mapping statistics (mapped reads, reference genome coverage, average depth), SNP calling, phylogenetic inference (tree congruence and topology) as well as parameters of interest from an evolutionary perspective such as the inference of natural selection and recombination rates (Fig 1). Particular emphasis has been given to the effects of reference selection that result in misleading epidemiological inferences.

## Results

### Selection of reference genomes

Complete genome sequences of five pathogenic bacterial species were downloaded from GenBank. These included *K. pneumoniae* (270 genomes), *L. pneumophila* (91 genomes), *N. gonorrhoeae* (15 genomes), *P. aeruginosa* (150 genomes) and *S. marcescens* (39 genomes). Only one strain from *P. aeruginosa* (KU, accession number CP014210.1) was discarded because of low assembly quality (32% of ambiguous positions). We built a ML core genome tree showing the phylogenetic relationships between the available assemblies for each species (S1 Fig). Based on this phylogenetic information and the strains commonly used in the literature, we selected 8 reference genomes for *K. pneumoniae*, 7 for *L. pneumophila*, 3 for *N. gonorrhoeae*, 6 for *P. aeruginosa* and 4 for *S. marcescens* (S1 Table), including the NCBI reference genome of each species. The strains 342 and AR_0080 (*K. pneumoniae*), and U8W and Lansing 3 (two *L. pneumophila* strains not included in subsp. *pneumophila*), and PA7 (a known 'taxonomic outlier' of *P. aeruginosa*) showed ANI values <95% in pairwise comparisons with the remaining selected references (S2 Table) and long branches separating them from the other references in their corresponding phylogenies (S1 File).

*In silico* MLST typing was performed for all the reference genomes except those of *S. marcescens*. The only cases of shared STs were found in strains HS09565, HS102438 and NTUH-K2044 of *K. pneumoniae* (ST 23), and in strains 32867 and CAV1761 of *N. gonorrhoeae* (ST 1901).

### Mapping to different references

We randomly sampled 20 isolates from different whole-genome sequencing data sets of the five bacterial species (S3 Table). Next, filtered and trimmed paired-end reads of each isolate were mapped to each reference genome from the same species. We computed different parameters for each mapping (S4 Table). The proportion of mapped reads and coverage of the reference genome (i.e., the percentage of reference genome covered by the aligned reads) showed variability depending on the reference used for mapping (Figs 2 and 3). Both parameters followed a roughly similar trend, as they presumably depend on the genetic distance between

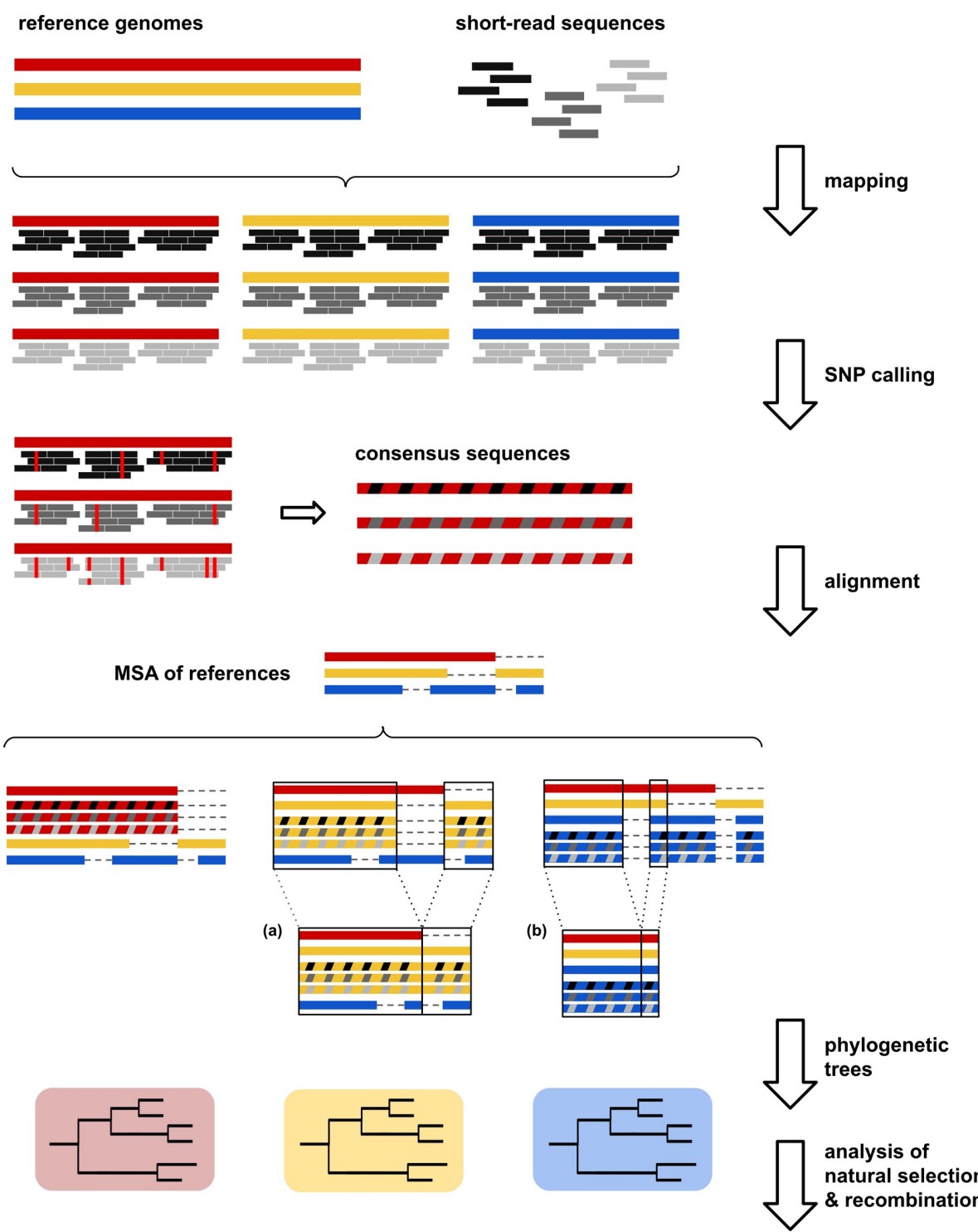

**Fig 1. Overview of the workflow used.** For each species, we selected different (3–8) publicly available closed whole-genome sequences as references and 20 sets of short-reads from whole-genome sequencing projects. Reads were mapped to each selected reference genome per species and consensus sequences were obtained from quality SNPs of each mapping. Consensus sequences from the mappings to the same reference genome were added to the MSA of all references of each species. For the analysis of each MSA, (a) we considered only those genome regions present in the reference used for mapping and (b) we obtained a 'core' MSA by removing all the regions absent from any of the reference sequences. Finally, we studied the impact of reference choice on the ML trees inferred from each MSA, recombination rates calculated on 'core' MSAs and dN/dS ratios calculated considering only coding sequences.

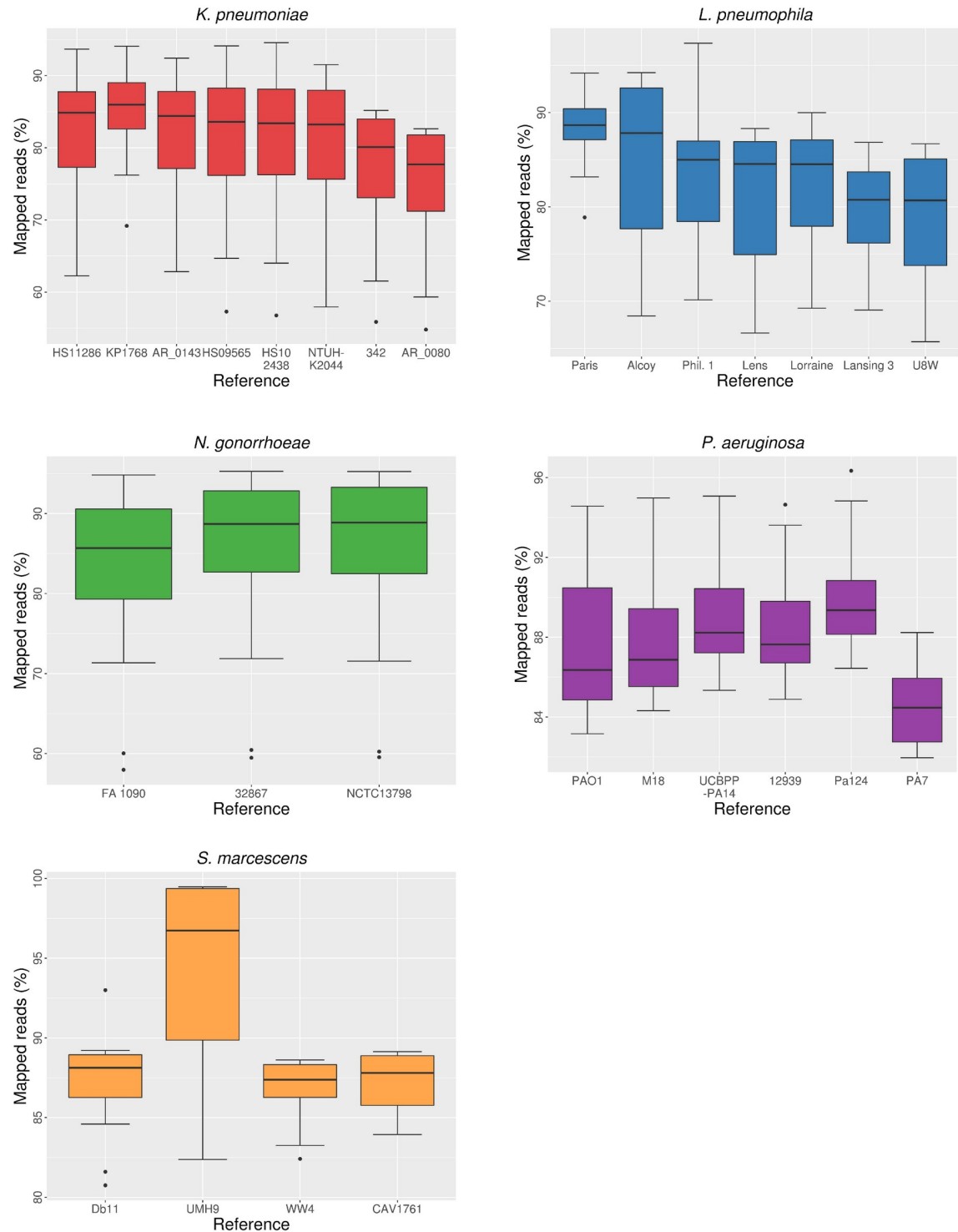

**Fig 2. Distribution of proportion of mapped reads depending on reference choice.**

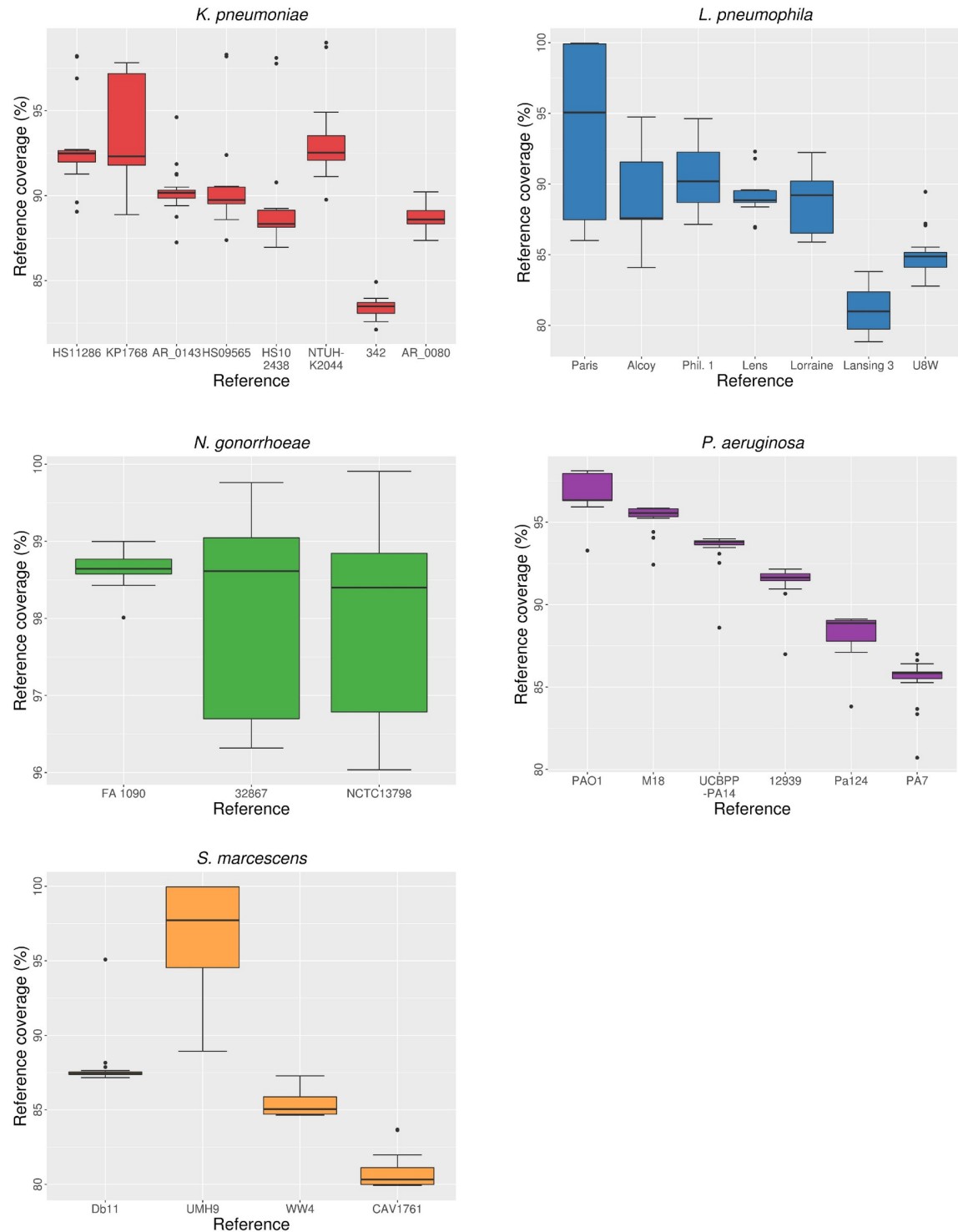

**Fig 3. Distribution of coverage of the reference genome depending on reference choice.**

**Table 1. Proportion of significant (P<0.05) comparisons depending on reference choice.**

| Species | Comparisons | Proportion (%) of significant comparisons | | | | |
|---------|-------------|-----------------|-----------------|-------|------|--------|
| | | Mapped reads[a] | Genome coverage[a] | SNPs[a] | ρ[b] | dN/dS[a] |
| K. pneumoniae | 28 | 7.1 | 75.0 | 53.6 | 42.9 | 60.7 |
| L. pneumophila | 21 | 19.0 | 52.4 | 95.2 | 23.8 | 47.6 |
| N. gonorrhoeae | 3 | 0.0 | 0.0 | 66.7 | 0.0 | 66.7 |
| P. aeruginosa | 15 | 26.7 | 93.3 | 86.7 | 73.3 | 53.3 |
| S. marcescens | 6 | 50.0 | 100 | 83.3 | 83.3 | 83.3 |

[a] Pairwise Bonferroni-corrected Wilcoxon tests.

[b] Pairwise Kolmogorov-Smirnov tests.

isolates and reference genomes. Moreover, we observed overlaps between the values obtained from mappings of the same isolates against different reference sequences in the five species. In most cases, the lowest median values were obtained in the alignments against the most genetically distant reference genomes (see 'Selected isolates and reference genomes'). However, the largest gap between median values depending on reference choice was found in the *S. marcescens* data set: the alignments to the outbreak-related reference UMH9 showed a high proportion of mapped reads (96.7%) and genome coverage (97.7%), whereas the alignment against the remaining references resulted in median values lower than 89% for both parameters. This was probably due to the high proportion of mapped reads and genome coverage resulting from mappings of outbreak isolates against a very close reference genome. Differences in both parameters were found to be significant (Kruskal-Wallis, P < 0.05) depending on the reference used for mapping in all species but *N. gonorrhoeae*. In the case of genome coverage, most pairwise comparisons (50%-100% in the four species) were found to be significant (Wilcoxon, P < 0.05), whereas the number of significant comparisons was lower for the proportion of mapped reads (Table 1). For example, in the case of *K. pneumoniae*, only 2 (out of 28) comparisons, involving the most genetically divergent reference genomes, showed significant differences in the proportion of mapped reads.

The average coverage depth (i.e., mean number of reads covering each position of the reference genome) was only slightly affected by reference choice (Fig 4 and S4 Table). Its effect was noticeable when reads were mapped to the most divergent reference genomes of the different species, as in the previous parameters. However, the average depth seemed to be more dependent on other factors such as the total number of reads (sequencing coverage) of the isolates rather than on the genetic distance to the reference genome. One such example is isolate NG-VH-50 (*N. gonorrhoeae*), which had a low total number of reads and also showed low average depth values regardless the reference selected for mapping (S5 Table). Differences in this parameter depending on the reference used for mapping were found to be non-significant in all the species, according to Kruskal-Wallis tests.

## SNP calling

SNPs were called and quality-filtered from the different mappings to each reference of the five species. The number of quality SNPs showed high variability depending on the reference used. Overlapping ranges of the number of called SNPs were found when comparing the results of the same isolates aligned to different reference sequences (Fig 5). Thus, considering that the number of SNPs between sequences is directly related to their genetic distance, SNP-calling results reflect genetic heterogeneity among isolates selected from the same species, as individual isolates showed different genetic relatedness to the different references.

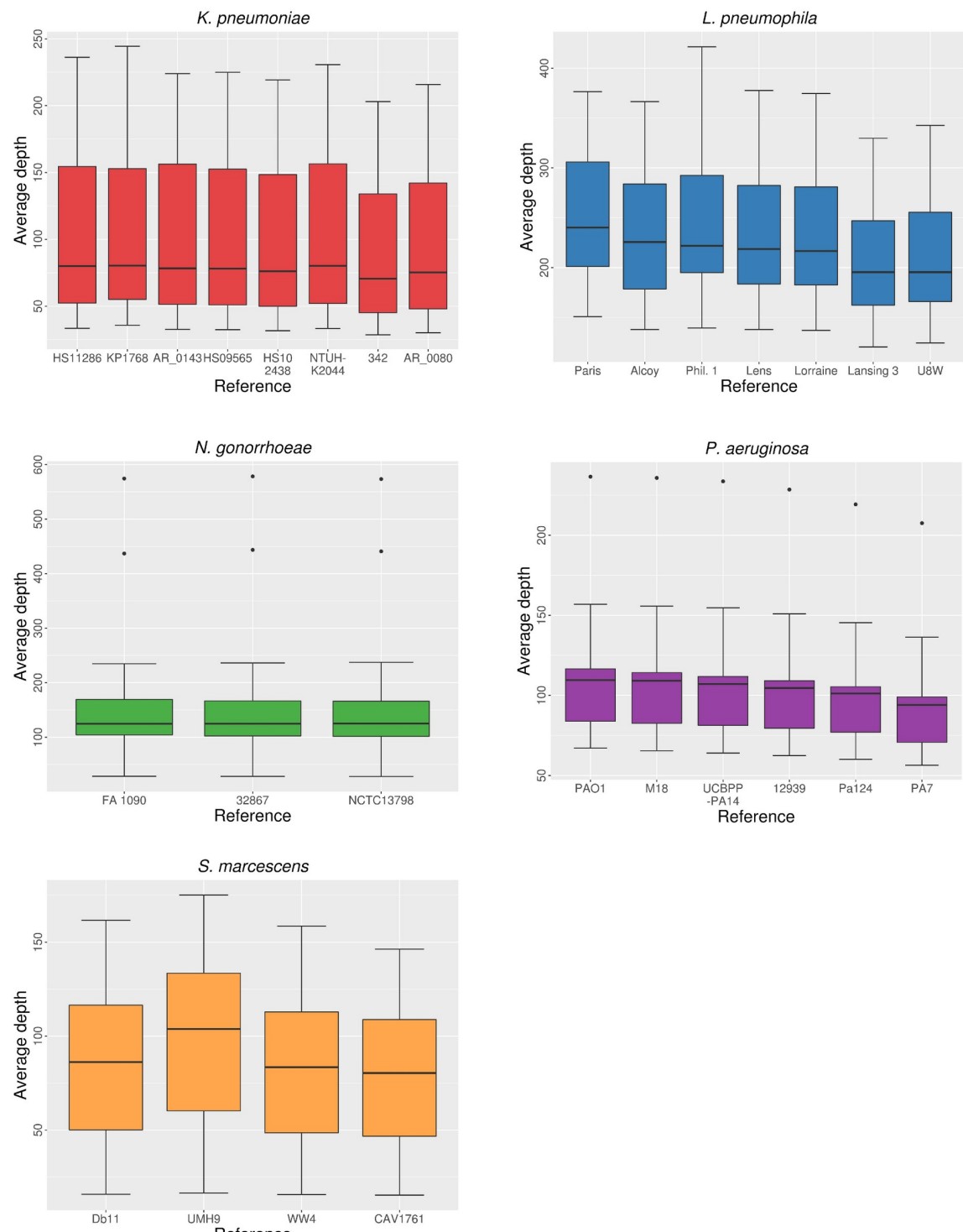

**Fig 4. Distribution of the average depth depending on reference choice.**

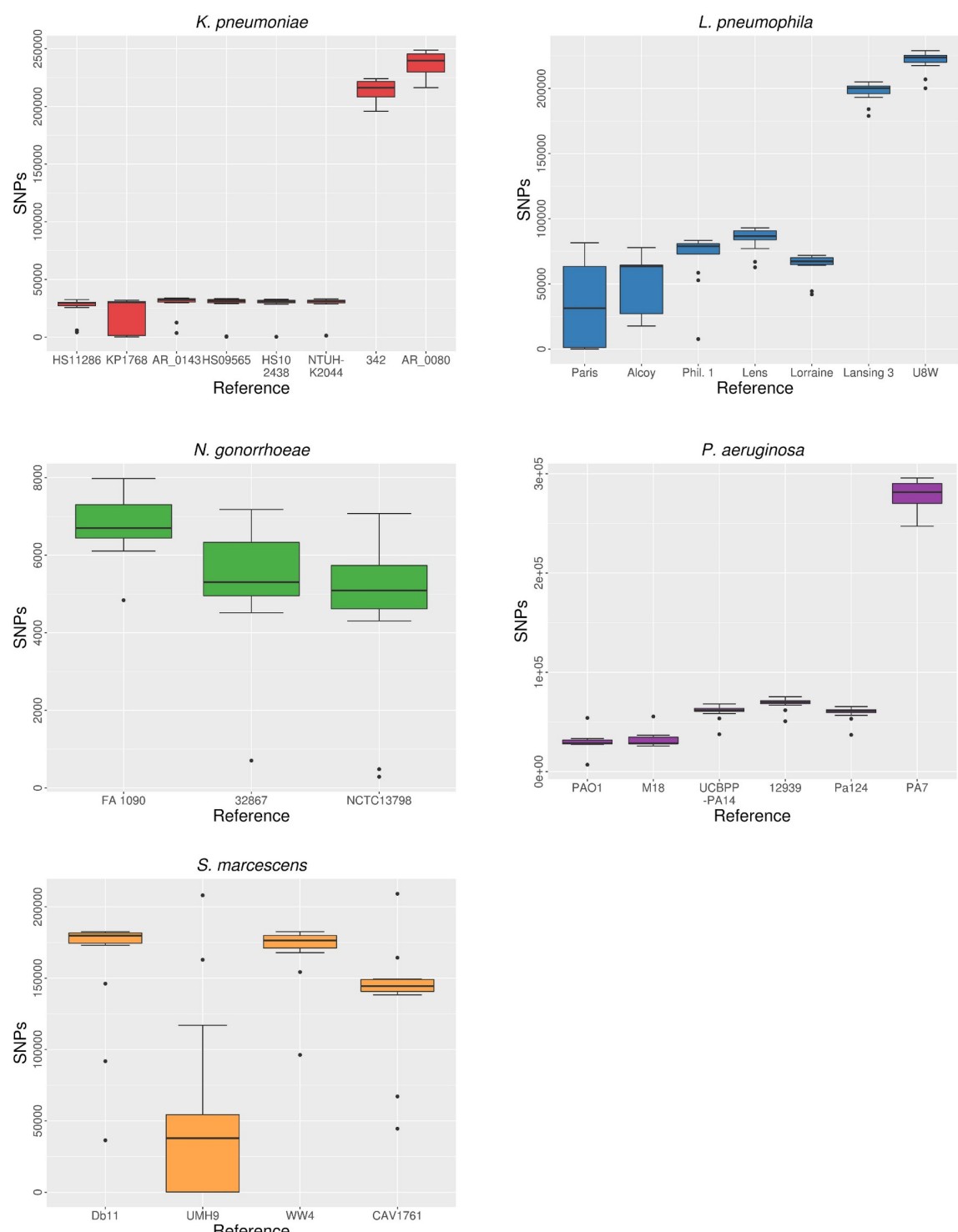

**Fig 5. Distribution of the number of SNPs depending on reference choice.**

**Table 2. Descriptive statistics of topological distances per species.**

| Species | Matching clusters | | | | | Robinson-Foulds clusters | | | | |
|---|---|---|---|---|---|---|---|---|---|---|
| | Mean | Median | SD | Min | Max | Mean | Median | SD | Min | Max |
| *K. pneumoniae* | 57.7 | 49 | 34.4 | 0 | 99 | 12.4 | 11 | 6.2 | 0 | 20 |
| *L. pneumophila* | 43.9 | 42 | 16.0 | 5 | 67 | 9.7 | 11 | 3.5 | 1 | 14 |
| *N. gonorrhoeae* | 40.0 | 44 | 13.5 | 25 | 51 | 8.7 | 10 | 3.2 | 5 | 11 |
| *P. aeruginosa* | 49.9 | 47 | 16.1 | 25 | 80 | 12.3 | 12 | 4.2 | 7 | 19 |
| *S. marcescens* | 31.3 | 29.5 | 7.9 | 21 | 43 | 6.5 | 6.5 | 2.9 | 3 | 10 |

An overall inverse relationship between SNP count and the previously discussed alignment parameters (mapped reads and genome coverage) was also observed (see Figs 2, 3 and 5). This implies that, in most cases, more SNP calls were expected in alignments with a lower proportion of mapped reads and genome coverage (which is roughly indicative of a worse performance of the read mapping process).

A relationship between the genetic distance of isolates to the reference sequence and the total number of SNPs called was clearly observed in the alignments against the most distant reference genomes of *K. pneumoniae*, *L. pneumoniae* and *P. aeruginosa*. These sequences, whose distances to all the isolates were expected to be high, showed SNP counts one order of magnitude larger than to other reference sequences (S4 Table).

In the case of *S. marcescens*, the alignments to strain UMH9 resulted in significantly fewer SNP calls when compared to mappings against the remaining reference sequences. This is explained by the presence of nearly identical isolates (outbreak isolates) to strain UMH9 (<160 SNPs detected). A similar case was found in *L. pneumophila* isolates 28HGV and 91HGV, which appeared to be nearly identical to the reference strain Paris, as less than 100 SNPs were detected in their respective mappings to this sequence. In all the species, most comparisons (53%-95%) between called SNPs from mappings against different references were significant (Wilcoxon, $P < 0.05$) (Table 1).

## Phylogenetic analyses and tree comparisons

We obtained a collection of MSAs including the same isolates and reference sequences, but differing only in the reference used for mapping by removing the regions absent in each mapping reference. We also obtained a 'core' genome MSA by removing simultaneously all the regions absent from any of the reference genomes for each species. Then, ML trees were inferred from each MSA. Due to methodology used to obtain the MSAs, the comparison between phylogenies strictly implies assessing the impact of reference selection.

Firstly, we quantified the topological distances between phylogenetic trees from each species with Robinson-Foulds clusters (RF) and matching clusters (MC) metrics. Tree distances spanned a variable range of values depending on the species (Table 2 and S6 Table). The normalized values of both metrics for the same tree comparisons were not equal (in most cases) but followed a similar global trend (Fig 6).

The comparisons involving phylogenies that include sequences mapped to the most divergent reference genomes of *K. pneumoniae* and *P. aeruginosa* showed the largest distance values. However, in most cases there was not a straightforward relationship between the genetic distance to the reference genomes and the topological distance between the corresponding trees (Fig 7). For example, *K. pneumoniae* trees using sequences from mappings to strains 342 and AR_0080 showed an identical topology (RF = 0, MC = 0), despite the ANI value between these references was <94%.

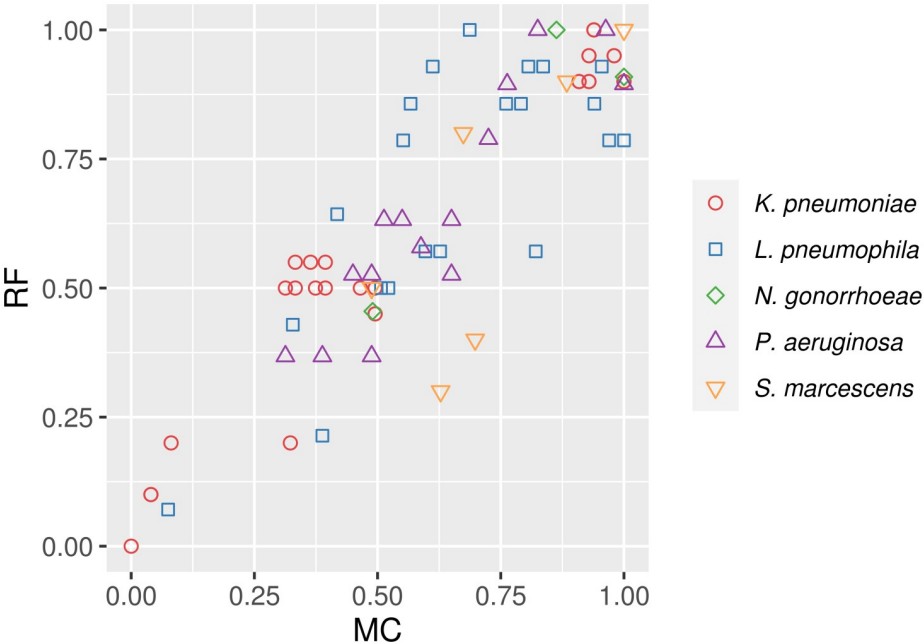

**Fig 6. Comparison of Robinson-Foulds (RF) and matching clusters (MC) normalized distances calculated between trees from the same species.**

The congruence between different tree topologies was rejected in most comparisons by ELW tests (Table 3). The few cases in which congruence was not rejected could be explained by the close phylogenetic relationship between the reference genomes involved.

Finally, in order to assess in detail the effects of reference selection on phylogenetic inference, trees from the same species were compared qualitatively. Changes in the phylogenetic

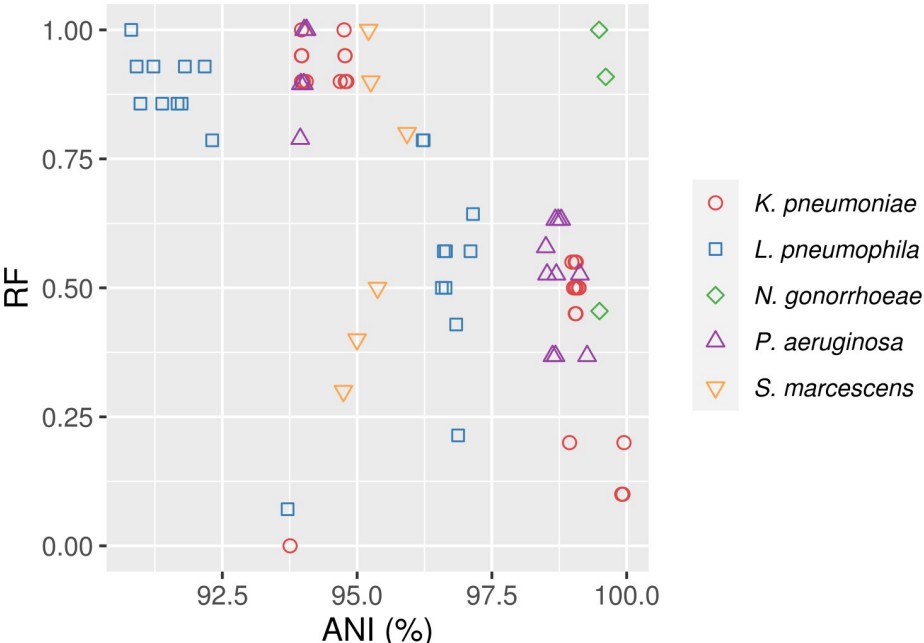

**Fig 7. Comparison of RF distances against ANI calculated between the reference genomes selected for each species.**

**Table 3. Congruent comparisons according to ELW test.** All the other pairwise comparisons were not congruent (P<0.05).

| Species | Reference | Congruent pair |
|---|---|---|
| *K. pneumoniae* | HS09565 | HS09565, NTUH-K2044 |
| | HS102438 | HS102438, NTUH-K2044 |
| | NTUH-K2044 | NTUH-K2044, HS09565 |
| | 342 | 342, AR_0080 |
| | AR_0080 | AR_0080, 342 |
| *L. pneumophila* | Lansing 3 | Lansing 3, U8W |

relationships were found when using different reference sequences in almost all cases except for two identical topologies. In some cases, the changes only affected branches in clades including closely related isolates (Fig 8A and 8B), while others implied more profound changes in the resulting topologies. Moreover, the alignments against a single reference genome seemed to underestimate the genetic distance between the consensus sequences of the isolates and the

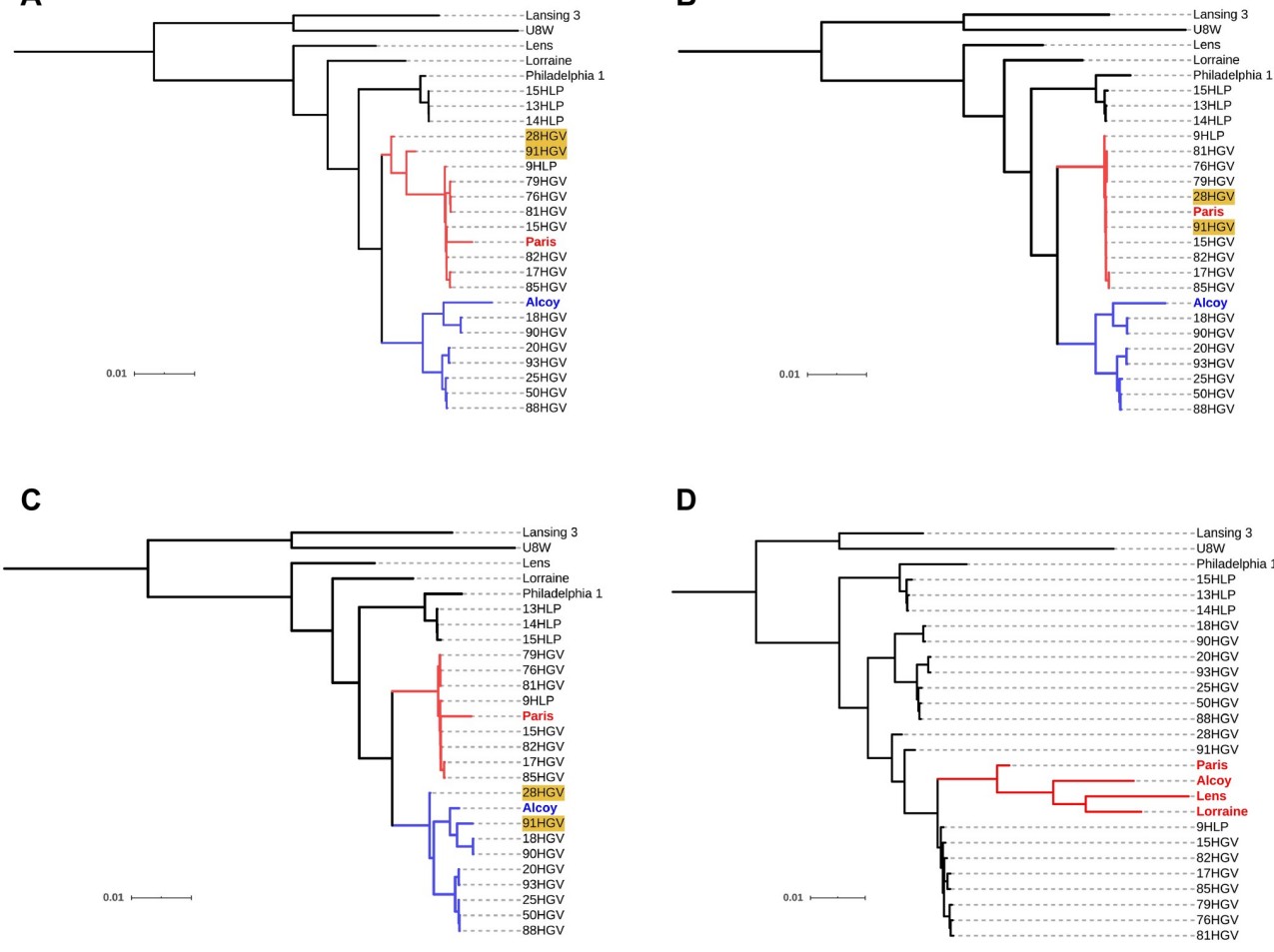

**Fig 8. Impact of reference choice on phylogenetic trees of *L. pneumophila*.** ML trees included the selected reference sequences of *L. pneumophila* and the consensus sequences obtained from mappings against strains (A) Philadelphia 1, (B) Paris, (C) Alcoy and (D) Lansing 3. Clusters of isolates related with references Paris (red) and Alcoy (blue) are coloured in the first three phylogenies. Isolates 28HGV and 91HGV (highlighted in yellow) were placed in different clades in the trees when using references Paris and Alcoy. Clade of references resulting from using Lansing 3 as reference genome is coloured in red.

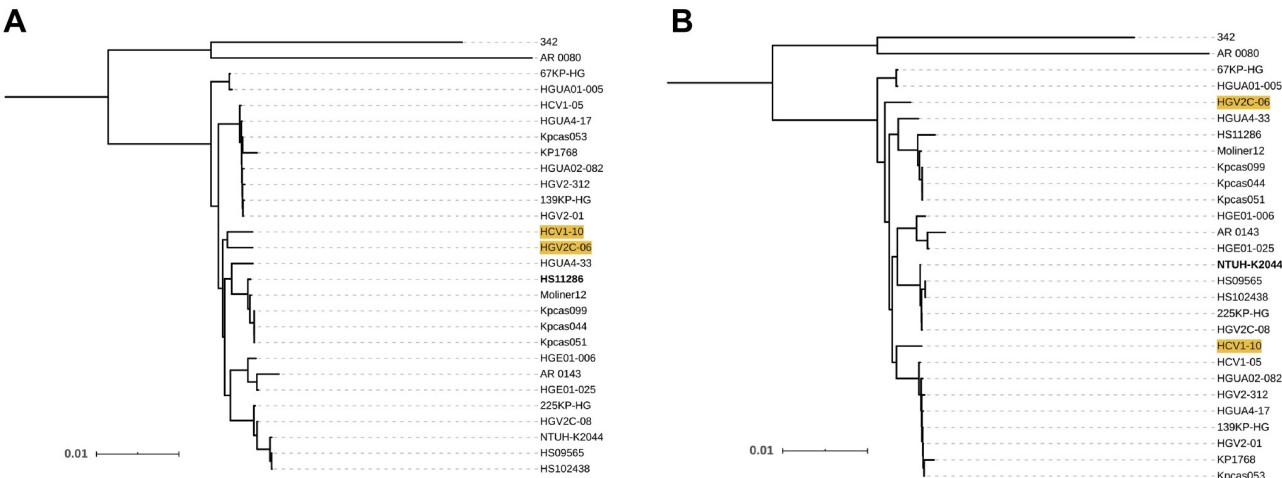

**Fig 9. Impact of reference choice on phylogenetic trees of *K. pneumoniae*.** ML trees included the selected reference sequences from *K. pneumoniae* and the consensus sequences obtained from mappings against strains (A) HS11286 and (B) NTUH-K2044. Isolates HGV2C-06 and HCV1-10 (yellow) changed their placement depending on reference choice.

reference sequence. Branch lengths were thus shortened between the leaves involved. In some extreme cases (when mapping to genetically distant genomes 342, AR_0080 [*K. pneumoniae*], Lansing 3, U8W [*L. pneumophila*] and PA7 [*P. aeruginosa*]), this 'attraction' effect led to the clustering of reference genomes not used as references for mapping in a single clade, regardless their genetic distance to the isolates (Fig 8D). These differences were also observed when only the core genome was used to obtain the phylogenetic tree (S2 File). Additional species-specific differences are described next.

**K. pneumoniae.** The topologies inferred with KP1768, NTUH-K2044, HS09565 and HS102438 as reference sequences revealed the same phylogenetic relationships between clusters of isolates, although there were some differences within clusters depending on the reference used for the MSA. Isolates HGV2C-06 and HCV1-10 (not associated with any of these clusters) changed their placement in the topologies with HS11286 and AR_0143 as reference sequences (Fig 9). The tree topologies using 342 and AR_0080 as reference genomes were identical and markedly different to the phylogenies derived with the other reference strains (S3 File).

**L. pneumophila.** The tree topologies using Lansing 3 and U8W as reference genomes were the most similar ones for this species (RF = 1, MC = 5) despite the large genetic distance between these sequences (ANI < 94%). Their topology was markedly different from the remaining topologies, where isolates grouped in three clades associated with reference genomes Paris, Alcoy and Philadelphia 1, respectively (see Fig 8 and S3 File). Notably, because of the epidemiological implications discussed below, isolates 28HGV and 91HGV were included in the Alcoy clade only when mapped to this reference genome (Fig 8C), whereas in all other cases (excluding U8W and Lansing 3) the isolates grouped with the Paris strain.

**N. gonorrhoeae.** The most similar topologies resulted from using FA 1090 and 32867 as reference genomes, despite that 32867 and NCTC13798 had larger ANI values. Three clades of isolates could be identified in all the phylogenies. However, those isolates not included in any of these clusters changed their position in the tree when using NCTC13798 as reference sequence in comparison with the two other trees. As an exception, isolate NG-VH-50 always grouped close to the reference sequence it was mapped to (S3 File). This artifact was due to the low total number of reads obtained in sequencing this strain.

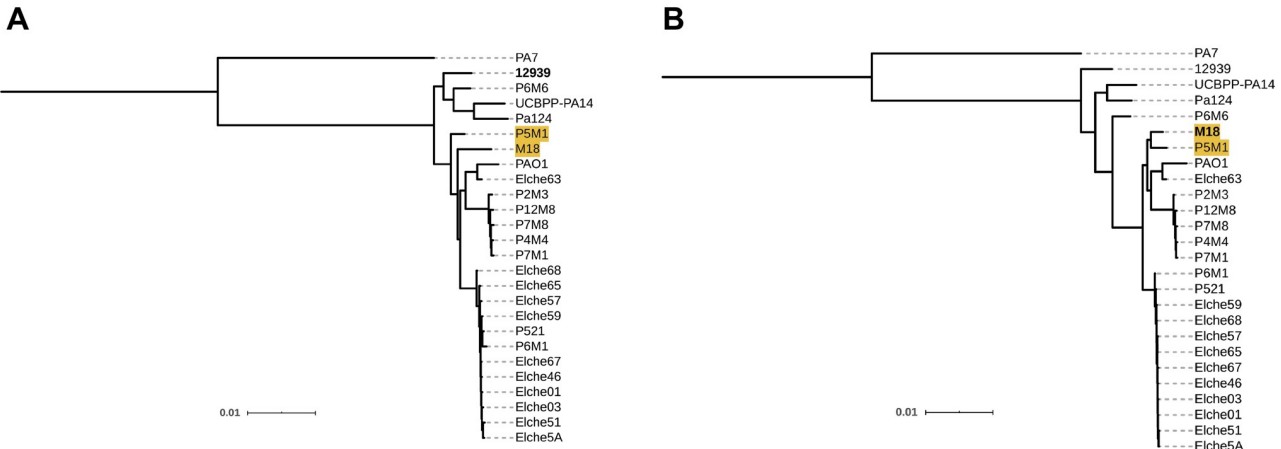

**Fig 10. Impact of reference choice on phylogenetic trees of *P. aeruginosa*.** ML trees included the selected reference sequences of *P. aeruginosa* and the consensus sequences obtained from mappings against strains (A) M18 and (B) 12939. Reference M18 and isolate P5M1 (yellow) alter their phylogenetic relationships depending on reference choice.

**P. aeruginosa.** Three clades were clearly identified in all the trees, with the exception of the one inferred using PA7 as reference sequence. In this tree, PA7 was placed in a cluster of isolates, whereas the remaining reference sequences clustered together (S3 File). The main topological differences depending on the reference were: (a) the placement of reference genome M18 and the isolate P5M1 in the tree, and (b) the phylogenetic relationships within the clade of reference genomes and P6M6, where the sequence chosen as reference for mapping occupied a basal position in the clade (Fig 10).

**S. marcescens.** Outbreak isolates grouped with strain UMH9 in all the trees. Branch lengths within this clade were practically null when UMH9 was used as the reference sequence, but these lengths increased when other reference sequences were used (Fig 11). As expected, the control isolate SMElx20 grouped with its closest reference (Db11) in all the cases. The phylogenetic relationships between reference genomes, isolates and clades changed depending on the reference used. The reference genome WW4 grouped with isolate CNH62 in all the topologies except when this strain was used as reference (S3 File).

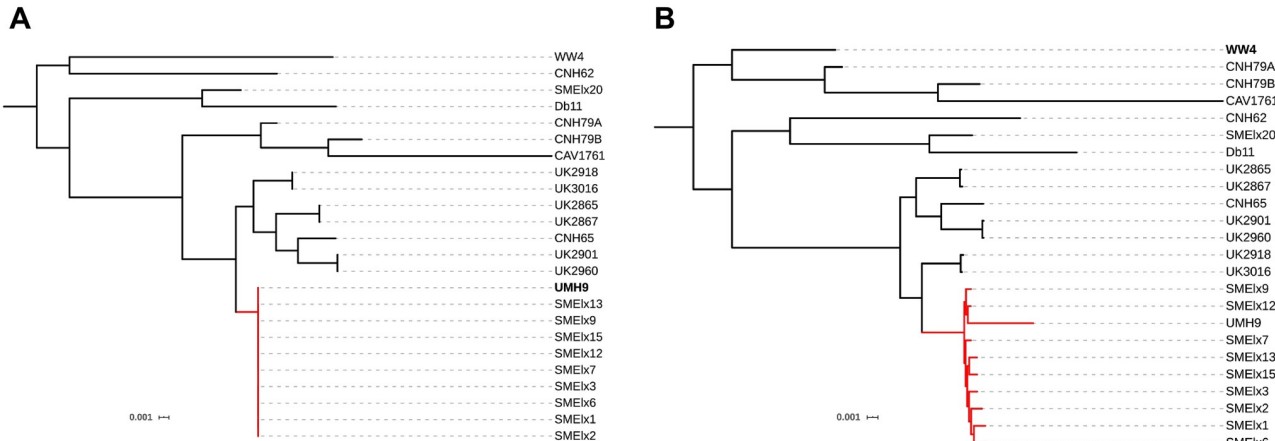

**Fig 11. Impact of reference choice on phylogenetic trees of *S.marcescens*.** ML trees included the selected reference sequences from *S. marcescens* and the consensus sequences calculated from alignments against strains (A) UMH9 and (B) WW4. Outbreak clade is shown in red.

## Distribution of recombination rates

Population recombination rates ($\rho$) were computed for 1000 bp sliding windows of the MSAs (S4 Table) and the corresponding distributions were compared. Those regions that were not present in all the sequences of a species were removed from the alignments for these analyses.

Overall, the distributions of recombination rates were very similar regardless the reference genome used in each case. However, relevant differences in some peaks were found in different MSAs from the same species. For example, the MSAs built with 32867 or NCTC13798 (*N. gonorrhoeae*) as reference sequences showed at least two clearly observable peaks that were absent when FA 1090 was the reference (Fig 12).

The number of significant pairwise comparisons between distributions of recombination rates (Kolmogorov-Smirnov, P < 0.05) differed widely depending on the species. While none of the comparisons between distributions of *N. gonorrhoeae* sequences showed significant results (although, as described previously, relevant differences were found), almost all *S. marcescens* estimated distributions were found to be significantly different (83.3%) (Table 1). In most cases, the significance of the comparisons between recombination rates could be explained by the phylogenetic relationships among the reference genomes. For example, the comparisons involving the most distant reference sequences of *K. pneumoniae*, *L. pneumophila* and *P. aeruginosa* showed significant differences, with the exception of the mutual comparisons between U8W and Lansing 3 (*L. pneumophila*), as well as AR_0080 and 342 (*K. pneumoniae*). Moreover, the significant comparisons in *P. aeruginosa* roughly reflected genetic distances between reference sequences, because using phylogenetically close reference sequences (M18 and PAO1 or UCBPP-PA14, Pa124 and 12939) resulted in non-significant differences between recombination rate distributions. In the case of *S. marcescens*, generalized significant comparisons could reflect nearly homogeneous divergence among the four reference genomes (S1 File).

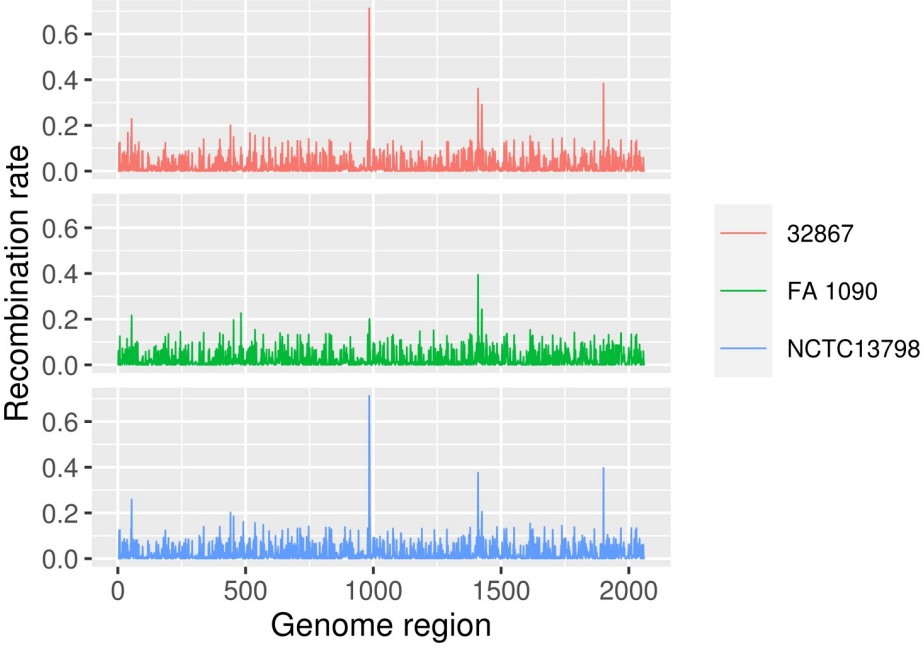

**Fig 12. Recombination rate distribution depending on reference choice between 'core' MSAs including sequences from *N. gonorrhoeae*.**

## Analysis of natural selection

Changes in the ratio ω (= d$N$/d$S$) due to reference choice could affect inferences on how natural selection has acted throughout the genome. This parameter was estimated in pairwise comparisons between concatenated CDS extracted from consensus sequences obtained from the mappings (S4 Table).

In all cases, the d$N$/d$S$ values computed for each gene were <1. Differences in d$N$/d$S$ depending on the reference used (Fig 13) were significant (Kruskal-Wallis, P < 0.05) for all the species. The proportion of significant pairwise comparisons (Wilcoxon, P < 0.05) depended on the species, ranging from 47.7% (*L. pneumophila*) to 83.3% (*S. marcescens*) (Table 1). In contrast with the results obtained in the parameters discussed previously, some of the comparisons involving the most genetically distant reference genomes (e.g., 342 strain of *K. pneumoniae*) as mapping references were not significant. Therefore, in this case it is difficult to explain the variability of ω based on the genetic distances between reference sequences for most species. *N. gonorrhoeae* could be treated as an exception, because the comparisons involving the reference strain FA 1090 (the most genetically distinct one) were the only significant ones. These differences were also observed when only the core genome was used to compute ω.

## Discussion

The impact of using different reference sequences for mapping NGS data sets has been studied previously in clinically relevant bacteria such as *Escherichia coli* [22], *Salmonella enterica* [26], *Listeria monocytogenes* [23,24,28,42] or *Mycobacterium tuberculosis* [25,28], as well as in eukaryotes [21,43,44], including *Homo sapiens* [45]. However, a systematic analysis of the evolutionary and epidemiological implications of reference choice, encompassing different bacterial species and diverse reference genomes is still missing. This work has been aimed at filling this gap. Indeed, in some cases, reference selection analysis is incidental, spanning a restricted number of reference sequences [46]. Among the species included in this work, the influence of reference diversity on SNP calling has been previously assessed in *K. pneumoniae* and *N. gonorrhoeae* [28], whereas *L. pneumophila*, *P. aeruginosa* (both showing high genomic variability [33,35]) and *S. marcescens* have not been studied under this perspective.

Statistics on raw mapping data such as the proportion of mapped reads and the coverage of the reference genome can provide preliminary information on the effect of reference choice and its effects on subsequent analyses, because these parameters reflect the performance of read alignment. As suggested previously, the genetic distance between short-read data and the reference genome is directly related to incorrect read alignment and unmapped reads due to mismatches between the sequence of the reads and the homologous positions in the reference [19,20,22]. This is also confirmed by our results on read alignment statistics. The percentage of the reference genome covered by mapped reads may be affected not only by genetic differences in homologous regions, but also by the presence of strain-specific genomic regions [21], because genes absent in the reference genome are expected to be lost during the mapping and in the subsequent multiple alignment. Moreover, as proposed by Lee and Behr [25], there might exist a coverage threshold beyond which subsequent phylogenetic analyses would be strongly affected, thus reducing the accuracy of evolutionary and epidemiological inferences derived from such inaccurate mappings.

The effect of sequencing coverage of the isolates on mapping seems to be generally independent of reference choice, as shown by the values of average coverage depth obtained in this study. Similarly to Pightling *et al.* [23], we have not observed any relationship between sequencing coverage and other variables during HTS data processing. However, as shown by one *N. gonorrhoeae* isolate (NG-VH-50), the reference mapping approach could strongly

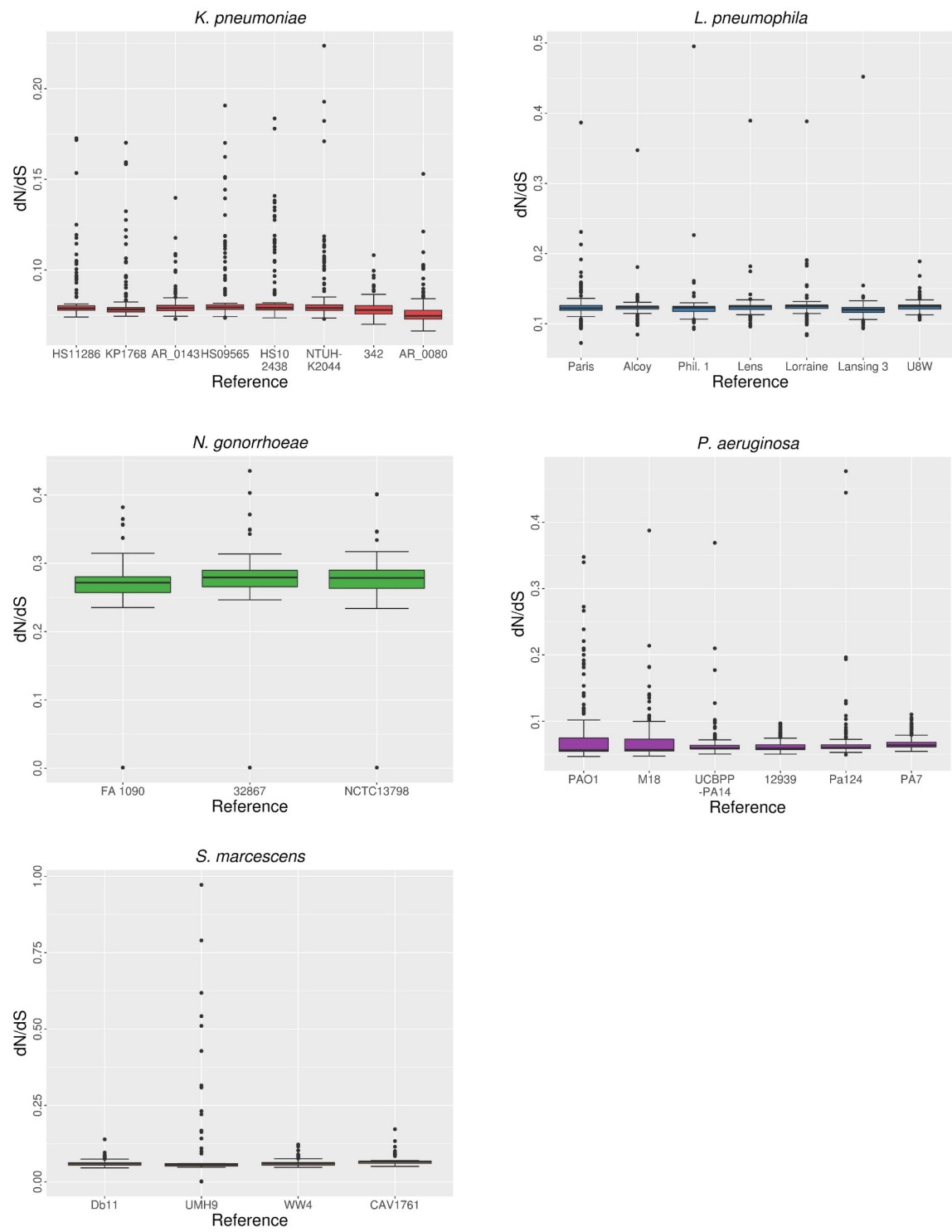

**Fig 13. Distribution of dN/dS depending on reference choice.**

underestimate the genetic distance between the assembly of the genome of a particular isolate and that of the reference genome below a certain threshold of total reads, thus affecting subsequent phylogenetic inferences.

Benchmarking of SNP calling performance for HTS data seems to be more common compared to other steps of genomic analyses [27,47–54]. Although most of these works are focused on assessing the effect of the selected pipeline (and its underlying algorithm), the use of different reference sequences has also been identified as a potential source of biases that could interact with other variables of the pipeline such as selection of the variant caller and read alignment software [23,24,28]. The number of SNPs is often used as a criterion for defining clusters of epidemiologically related isolates [55]. Our results confirm the existence of a systematic and significant influence of reference choice on the number of identified SNPs in all the species analyzed. They also reflect the correlation between genetic distance of isolates to the reference genome and the number of called variants which, as highlighted in previous studies, could be associated with the increase of false positives when the precision of SNP calling decreases [23,28,42]. However, the increase in false positive rates in variant detection may be affected not only by distance between query and reference genomes (measured as SNP distance), but also by genomic architecture or assembly quality [56]. This suggests that multiple aspects should be considered when choosing a reference for mapping, aside from SNP distance between reads and reference genome. Furthermore, SNPs could also provide information about genomic heterogeneity of the data sets, as overlapping ranges in the number of SNPs called depending on the reference reflects that different isolates are closely related to different mapping references.

Recovering phylogenetic relationships between organisms or strains within a species represents an essential procedure in evolutionary and epidemiological studies. Biases in how and how many SNPs are called as well as in the gene content of the final assemblies due to reference choice could affect phylogenetic inferences [47]. The overall negative results obtained in congruence tests also reflect the existence of a systematic effect of reference choice on tree topologies: the only statistically concordant comparisons (6 out of 73) between topologies of the same species were found when references chosen for mapping were (a) closely related sequences (*K. pneumoniae* ST 23 strains), or (b) extremely distant sequences, showing ANI values close to the boundaries for species delimitation. The topologies resulting from using phylogenetically unrelated, extremely divergent genomes were mutually similar while, in contrast, generally showed high topological distance values when compared to trees built using non-extreme references. This kind of loss in tree resolution has already been observed (although limited to clonal bacteria [25]). In our case, it may be originated from a reduced proportion of shared gene content between isolates and extremely divergent sequences, along with the existence of barriers to recombination between populations, as the ability for recombination and its frequency is expected to decrease with genetic distance [57]. However, these differences were also observed when considering only the core genome. This suggests that the effect of the reference on phylogenetic inference is not only due to the presence/absence of genes in the accessory genome. It might be due also to differences in core genome sequences arising from biased/erroneous identification of variants.

The effect of reference choice on phylogenetic inferences is pervasive in these five species. However, despite the differences between topologies and even lack of congruence, these changes might not be necessarily associated with altered epidemiological inferences. A similar situation was studied by Usongo *et al.* [26] on a *S. enterica* epidemiological data set, in which two different topologies (RF = 24) were resolutive enough to distinguish different outbreak clusters. In the same way, Kaas et al. [58] built phylogenies with the same clusters regardless using a close or distant reference. In contrast, Abdelbary et al. [59] showed that overestimation

of the number of SNPs due to mapping against a distant reference in outbreak investigations could potentially affect phylogenetic reconstruction and lead to misinterpretation of the results, thus suggesting that using a de novo assembled outbreak-related sequence as reference (if a closer reference genome is not available) would improve mapping statistics and decrease SNPs false positive rates. Through the systematic analysis of mapping in different species and using a genetically diverse set of reference sequences, we have showed that both observations are compatible, and that the effects on phylogenies and epidemiological inferences depend on the references compared: the use of different reference sequences affects phylogenetic relationships between clades and even to the association of specific isolates to transmission clusters, thus potentially affecting epidemiological inferences. These alterations have been observed not only when using a distant reference, but even when mapping to phylogenetically related strains from the same non-clonal species as a reference, in contrast with previous studies in clonal bacteria [25] where differences in phylogenetic inference appeared when using reference genomes from close but different species. This is most obvious in the *L. pneumophila* data set, in which two isolates changed their positions and were placed in the same cluster of the reference sequence used for mapping, while the overall topology remained practically unchanged.

Differences between trees were quantified by topological distance metrics, reflecting, in most cases, lack of correlation between tree distances and genetic distances of the corresponding reference genomes. As suggested previously [22,27], when working with a genetically diverse set of isolates, it is impossible to select a single reference close to all of them, and single-reference mapping biases are expected to increase with genomic divergence. Therefore, these differences in tree topologies could be partially explained by the use of genetically heterogeneous data sets. Moreover, its impact on tree reconstruction may be alleviated by using multiple references simultaneously or a reference pangenome instead [22,60–64]. If data sets of isolates were homogenous (i.e., the isolates are equally close to the same reference) as the one employed by Lee and Behr [25], we would expect that read alignment performance and tree resolution would decrease as we select progressively distant reference genomes [23,24,28].

However, we could not ignore that the presence of recombination (particularly in highly recombinogenic species such as *K. pneumoniae* and *L. pneumophila*) could reduce accuracy in phylogenetic reconstruction [22], thus explaining to some extent the topological incongruence or the differences in branch lengths [65].

Selecting one reference or another for mapping can also affect the estimates of phylogenetic distance between isolates [22,26], which is reflected in the branch lengths of the trees. This is clearly illustrated by the phylogenetic analysis of the *S. marcescens* data set, which reveals that tree branches connecting outbreak isolates increased their lengths when consensus sequences were calculated from alignments using reference genomes that were phylogenetically unrelated to the isolates (different from strain UMH9). Similar findings were observed for *Listeria monocytogenes* sequences by Pightling *et al.* [23].

The development and increasing availability of high-throughput, whole-genome sequencing technologies have allowed assessing evolutionary rates and dynamics at the genome level which, in turn, contribute to a better understanding of emerging diseases and transmission patterns [66]. Therefore, the study of natural selection and recombination, frequent processes in bacteria [67], is relevant not only from an evolutionary point of view but also in its application to molecular epidemiology [68]. To our knowledge, the impact of reference selection on the inference of evolutionary parameters such as substitution and recombination rates at the genome level has not been explored previously. In this work, variations in d$N$/d$S$ and $\rho$ have been detected in all the species depending on the reference sequence used for mapping. This might have an effect in subsequent inferences on the action of natural selection and the detection of recombination events. Significant differences in $\rho$ seemed to be more strongly

correlated with the genetic distance between the genomes used as reference for mapping than d$N$/d$S$.

Short-read mapping of HTS data against a reference genome is a common approach in bacterial genomics. Our results show that the impact of selecting a single reference is pervasive in the genomic analyses of five different bacterial species, and likely in many others. All the parameters evaluated were affected by the usage of different reference sequences for mapping and, notably, alterations in phylogenetic trees modified in some cases the epidemiological inferences. Furthermore, working with heterogeneous sets of isolates seems to be a particularly challenging scenario for the selection of a single reference genome. Mapping simultaneously to multiple references or against a reference pangenome may alleviate the effect of reference choice. Moreover, when studying particular lineages or outbreaks, using de novo assembled, closely related references (i.e., sequences from the same clade) may also reduce this effect. However, the aim of this work is not to provide a solution on the reference selection bias on mapping but to make clear how deeply reference choice can affect subsequent analyses. Exploring the effects of different references is highly recommended, since it is difficult to unequivocally determine an optimal reference when working with non-simulated reads, further considering that available, high-quality reference genomes may not encompass the complete genomic diversity of a species. Besides, the diversity and uniqueness of each biological dataset impedes the elaboration of a generalizable guideline. Ultimately, inspecting the variability of the results of mapping against different references is an essential step to assess if conclusions are robust to reference choice and which of them are particularly sensitive to the use of specific references.

## Methods

The workflow used in this study is summarized in Fig 1.

### Selection of reference genomes

Closed whole-genome sequences of *K. pneumoniae*, *L. pneumophila*, *N. gonorrhoeae*, *P. aeruginosa* and *S. marcescens* available in June, 2018 were downloaded from NCBI GenBank [69] in fasta format. Plasmids were removed with seqtk v1.0 (https://github.com/lh3/seqtk) (subseq command). Genome sequences were annotated using Prokka v1.12 [70] (with default settings) and the set of intra-species co-orthologous genes was inferred using Proteinortho v5.11 [71] (option -p = blastn+). Coding sequences (CDS) of orthologous genes in each species were aligned with MAFFT v7.402 [72] (with default settings) and concatenated to obtain a CDS-coding core genome multiple sequence alignment (MSA) for each species.

A maximum-likelihood (ML) tree was inferred from each MSA with IQ-TREE v1.6.6 [73] using the GTR substitution model and 1000 fast bootstrap replicates [74]. After consideration of the core genome phylogenies (distance between strains and clusters) and the usage of different references in the literature, we selected a set of genomes to be employed as reference genomes for each species. The number of reference sequences selected was roughly proportional (≈10%) to the initial number of publicly available sequences from each species. In brief, we included (a) the NCBI reference genome of the species, (b) relevant or commonly used references for mapping, and (c) representative sequences of different lineages. Detailed information about the selected reference genomes is provided in S1 Table.

The selected reference genomes of each species were aligned with progressiveMauve v2.4 [75] and gaps were added to regions where homologous sequences were absent in any

genome in the alignment. The XMFA output alignment was converted into fasta format with xmfa2fasta.pl (https://github.com/kjolley/seq_scripts/blob/master/xmfa2fasta.pl).

To evaluate the genetic divergence between the selected reference sequences, we used three different procedures: (a) we built ML trees with IQ-TREE, as above, (b) we computed Average Nucleotide Identities [76] (ANIs) using FastANI v1.1 [77], and (c) we performed an *in silico* multi-locus sequence typing (MLST) using mlst v1.15.1 (https://github.com/tseemann/mlst) for *K. pneumoniae*, *N. gonorrhoeae* and *P. aeruginosa*; and using BLAST+ [78] and the EWGLI [79] database for *L. pneumophila*. This procedure was not used with *S. marcescens*.

## Selection of isolates for analysis

20 sets of short-reads from whole genome sequencing projects of the five species (S3 Table) were randomly selected (with the R [80] function sample_n) among those obtained in our laboratory and/or deposited at the SRA as detailed next. Sequences in our laboratory were obtained with Illumina MiSeq 300x2 paired-ends (*P. aeruginosa*) or NextSeq 150x2 paired-ends (the remaining species). The *K. pneumoniae* data set included isolates of 9 different STs obtained in a surveillance study of ESBL-producing strains in the Comunitat Valenciana (Spain). The *L. pneumophila* data set comprised isolates obtained from environmental surveillance at 2 hospitals of the Comunitat Valenciana. The *N. gonorrhoeae* data set includes isolates obtained in a surveillance study in different regions of Spain (Comunitat Valenciana, Madrid and Barcelona). The *P. aeruginosa* data set included isolates from 2 outbreaks detected in the Comunitat Valenciana. Finally, the *S. marcescens* data set included 9 almost identical outbreak isolates genetically close to strain UMH9, one isolate close to the reference of the species, Db11, and 10 unrelated isolates downloaded from the SRA repository.

## Quality control analysis and sequence read processing

The quality of the reads (before and after trimming and filtering) was assessed using FastQC v0.11.8 (https://www.bioinformatics.babraham.ac.uk/projects/fastqc/) and quality reports were merged with MultiQC v1.7 [81]. Illumina, Truseq and Nextera adapters were removed with cutadapt v1.18 [82]. Reads were trimmed and filtered using Prinseq-lite v0.20.4 [83]. 3'-end read positions with quality <20 were trimmed and reads with overall quality <20, >10% ambiguity content and total length <50 bp were removed.

## Mapping, variant calling and consensus sequences

Reads passing the above filters were mapped to each selected reference of each species using BWA MEM v0.7.17 [84] (with default settings). SAM files were converted to binary format (BAM), sorted and indexed with samtools v1.6 [85] (commands sort and index). Mapping statistics were obtained using samtools (commands flagstats and depth).

SNPs were identified in each alignment with samtools and bcftools v1.6 [86] (commands mpileup and call, respectively). Indels were excluded from the analysis (option—skip-variants indels). Remaining SNPs after filtering (quality >40, mapping quality [MQ] >30, depth >10 and under twice the average depth and distance of >10 pb to any indel) were counted with bcftools (command stats).

Consensus sequences were obtained from quality-filtered SNPs and the appropriate reference sequence using bcftools (command consensus) for every possible combination of isolates and reference genomes from the same species.

## Multiple sequence alignment of reference genomes and consensus sequences

The MSAs of the reference sequences from each species were used as 'backbones' on which the consensus sequences from the mappings to the same reference genome were added using a custom Python script. XMFA-formatted MSAs were converted to fasta format as described previously. Finally, for the analysis of each MSA we considered only those genome regions present in the reference genome, using a custom Python script to mask the absent regions from the global MSA. This procedure (see Fig 1) allowed us to obtain a collection of MSAs (one per each reference sequence) including the same isolates and reference genomes (per species), differing only in the reference sequence used for mapping. In addition, we also obtained a 'core' genome MSA by removing all the regions absent from any of the reference sequences.

## Analysis of natural selection

We explored the effect of reference choice on the inference of natural selection at the whole genome level by computing pairwise d$N$/d$S$ ratios with the PAML package 4.9i [87] between concatenated CDSs of consensus sequences that were built using the same reference. CDSs were extracted using coordinates of the corresponding reference obtained with Prokka (see 'Selection of reference genomes'). A custom Python script and the emboss package v6.6.0 [88] were used. We also computed pairwise d$N$/d$S$ values between consensus sequences considering only the core genome CDSs (i.e., shared by all the selected references from each species).

## Distribution of recombination rates

Population recombination rates ($\rho = 4N_e r$; where $N_e$ is the effective population size and $r$ is the recombination rate per base pair and generation) were estimated using LDJump [89] (with a window of 1000 pb) from the 'core' genome MSAs. The distributions of recombination rates along MSAs were compared for the different reference genomes of each species and were represented graphically with the R package ggplot2 [90].

## Comparisons of phylogenetic trees

ML trees were inferred from each MSA with IQ-TREE as described above, and visualized with iTOL v4 [91].

**Congruence tests.** We used expected likelihood weight (ELW) tests [92], as implemented in IQ-TREE, to assess the congruence between phylogenies that differed only in the genome chosen as mapping reference. The ELW test computes weights for each topology based on its likelihood given a MSA, with the total sum of weights being equal to 1 and higher weights assumed to be those best supported by the data. Decreasing weights are progressively collected to build a confidence set until their cumulative sum is equal to or higher than 0.95. At this point, the trees included in the confidence set are accepted as congruent.

**Topological distances.** Pairwise distances between tree topologies obtained with the different mapping references were assessed using TreeCmp v2.0 [93]. Robinson-Foulds [94] clusters (RF) and matching clusters [93] (MC) metrics were calculated for each comparison. The RF distance reflects the number of bipartitions differing between topologies, whereas the MC distance computes the minimal number of moves needed to convert a topology into another. Therefore, two identical topologies will receive a value equal to 0 with both metrics. Conversely, distance values will increase as the compared trees become more different.

**Qualitative comparison of trees.** Finally, a qualitative assessment of trees was performed in order to identify specific changes in the phylogenetic relationships between isolates due to

the choice of different reference genomes. Particularly, we focused on clustering of isolates and alterations that could affect epidemiological inferences (e.g., including/excluding one particular sample in an outbreak).

**Statistical analyses.** To study the effect of using different reference genomes on mapping statistics (proportion of mapped reads, genome coverage, average depth), number of called SNPs, and d$N$/d$S$ values, non-parametric Kruskal-Wallis [95] tests were performed with R 3.5 (function kruskal.test). If a Kruskal-Wallis test showed significant differences between groups (reference sequence), we performed pairwise Wilcoxon [96] tests with Bonferroni-corrected p-value for multiple comparisons (with the R function pairwise.wilcox.test) in order to identify significant differences between specific reference sequences.

Pairwise Kolmogorov-Smirnov [97] tests (R function pairwise_ks_test [https://github.com/netlify/NetlifyDS]), which compare observed distributions of data, were performed in order to identify significant differences in the distributions of recombination rates depending on the mapping reference.

## Supporting information

**S1 Fig. Core genome trees of the complete whole-genome sequences downloaded from GenBank.** The circles at the tips denote the sequence type (ST) of the different strains in the trees of the species with an MLST scheme available for *in-silico* typing. The black triangles denote the branches with bootstrap support values <70. (A) *K. pneumoniae*, (B) *L. pneumophila* and (C) *P. aeruginosa* trees were rooted on their corresponding longest branches. As all the branches connecting the different clades of (D) *S. marcescens* and (E) *N. gonorrhoeae* trees were approximately the equal length, they were rooted arbitrarily for a better visualization.
(PDF)

**S1 Table. Strains selected as references for mapping.**
(XLSX)

**S2 Table. ANI (%) calculated between the selected reference genomes.**
(XLSX)

**S3 Table. Isolates (short-read sequence data) selected for mapping.**
(XLSX)

**S4 Table. Summary statistics per reference and species.** Median, minimum and maximum values are shown.
(XLSX)

**S5 Table. Mapping and SNP statistics per reference and species.**
(XLSX)

**S6 Table. RF and MC distances.**
(XLSX)

**S1 File. Phylogenetic trees of the reference genomes selected for each species.**
(ZIP)

**S2 File. Core genome phylogenetic trees per reference and species.** Strain selected as reference for mapping in each tree is indicated in the corresponding newick file name.
(ZIP)

**S3 File. Phylogenetic trees per reference and species.** Strain selected as reference for mapping in each tree is indicated in the corresponding newick file name.
(ZIP)

## Author Contributions

**Conceptualization:** Beatriz Beamud, Fernando González-Candelas.

**Data curation:** Carlos Valiente-Mullor, Carlos Francés-Cuesta, Neris García-González, Lorena Mejía, Paula Ruiz-Hueso.

**Formal analysis:** Carlos Valiente-Mullor.

**Funding acquisition:** Fernando González-Candelas.

**Investigation:** Carlos Valiente-Mullor.

**Methodology:** Beatriz Beamud, Fernando González-Candelas.

**Project administration:** Fernando González-Candelas.

**Resources:** Carlos Francés-Cuesta, Neris García-González, Lorena Mejía, Paula Ruiz-Hueso, Fernando González-Candelas.

**Software:** Carlos Valiente-Mullor, Iván Ansari.

**Supervision:** Beatriz Beamud, Fernando González-Candelas.

**Validation:** Carlos Valiente-Mullor.

**Visualization:** Carlos Valiente-Mullor.

**Writing – original draft:** Carlos Valiente-Mullor.

**Writing – review & editing:** Beatriz Beamud, Fernando González-Candelas.

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
