## [Decision Letter · Decision Letter 0]

26 Aug 2020

Dear Dr. Beamud,

Thank you very much for submitting your manuscript "One is not enough: on the effects of reference genome for the mapping and subsequent analyses of short-reads" for consideration at PLOS Computational Biology.

As with all papers reviewed by the journal, your manuscript was reviewed by members of the editorial board and by several independent reviewers. In light of the reviews (below this email), we would like to invite the resubmission of a significantly-revised version that takes into account the reviewers' comments.

We cannot make any decision about publication until we have seen the revised manuscript and your response to the reviewers' comments. Your revised manuscript is also likely to be sent to reviewers for further evaluation.

Sincerely,

Kin Fai Au

Guest Editor

PLOS Computational Biology

Jian Ma

Deputy Editor

PLOS Computational Biology

Reviewer's Responses to Questions

**Comments to the Authors:**

Reviewer #1: This paper develops authors points out that the choice of a reference in microbial genomics analysis may represent some errors in the detection of SNPs and phylogenetic inference. The procedures are described clearly, and the evidences are convincing.

Comments:

1, I think the Fig. 13 is a very good summary of this paper. To let readers more easily to understand the whole paper, I suggest authors to refer to this figure in Introduction section and rename this as Fig. 1.

2, The code in Github does not include a description on how to use it. The code should contain a detailed manual to make it easier for users to use it.

Reviewer #2: This manuscript systematically analyzed the effects of reference selection in

short-reads data analysis, especially its impact in phylogeny reconstruction

and epidemiological inferences. I have two main concerns:

1. The novelty of this manuscript is rather limited, as similar work

has been done to analyze the effects of reference selection in SNP calling, etc.

2, There is no practical guidance on how to select references so that

the resulting biases can be reduced. This limits the impact of this manuscripts

in practical data analysis.

Despite the merits of this manuscript in quantitatively measuring the effects

of reference selection, I wouldn't suggest accepting it mainly due

to above two limitations.

**Have all data underlying the figures and results presented in the manuscript been provided?**

Reviewer #1: Yes

Reviewer #2: Yes

PLOS authors have the option to publish the peer review history of their article (what does this mean?). If published, this will include your full peer review and any attached files.

Reviewer #1: No

Reviewer #2: No
---

## [Decision Letter · Decision Letter 1]

9 Dec 2020

Dear Beamud,

Thank you very much for submitting your manuscript "One is not enough: on the effects of reference genome for the mapping and subsequent analyses of short-reads" for consideration at PLOS Computational Biology. As with all papers reviewed by the journal, your manuscript was reviewed by members of the editorial board and by several independent reviewers. The reviewers appreciated the attention to an important topic. Based on the reviews, we are likely to accept this manuscript for publication, providing that you modify the manuscript according to the review recommendations.

The authors should make minor revisions based on the reviewer's suggestion prior to be accepted for publication.

Sincerely,

Kin Fai Au

Guest Editor

PLOS Computational Biology

Jian Ma

Deputy Editor

PLOS Computational Biology

[LINK]

The authors should make minor revisions based on the reviewer's suggestion prior to be accepted for publication.

Reviewer's Responses to Questions

**Comments to the Authors:**

Reviewer #1: I have no further comments.

Reviewer #2: In the letter of response to reviewers, the authors made clear the novelty of

the work in evaluating the effects of reference selection in recombination,

natural selection inferences, and epidemiological analyses, and a new procedure

that guarantees only evaluating the effect of the reference. A GitHub

repo was well-documented which can help users to evaluate the efforts

of the reference with their own data. These addressed my main concerns;

I appreciate the authors' efforts.

Two minor concerns / suggestions:

1, I would suggest highlight the *unique* contribution of this work

in "Author summary" section, by adding/rephrasing the statements made in the letter,

for examples, "To our knowledge, this is the first work to systematically

examine the effect of different references for mapping on the inference of tree

topology as well as the impact on recombination and natural selection

inferences", and "The novelty of this work also relies on a procedure that

guarantees that we are evaluating only the effect of the reference".

2, Would it be possible to add a working (small) example to the GitHub repo~(for examples,

necessary files in references and reads folders)?

**Have all data underlying the figures and results presented in the manuscript been provided?**

Reviewer #1: Yes

Reviewer #2: Yes

PLOS authors have the option to publish the peer review history of their article (what does this mean?). If published, this will include your full peer review and any attached files.

Reviewer #1: No

Reviewer #2: No
---

## [Editor Report · Decision Letter 2]

5 Jan 2021

Dear Beamud,

We are pleased to inform you that your manuscript 'One is not enough: on the effects of reference genome for the mapping and subsequent analyses of short-reads' has been provisionally accepted for publication in PLOS Computational Biology.

Best regards,

Kin Fai Au

Guest Editor

PLOS Computational Biology

Jian Ma

Deputy Editor

PLOS Computational Biology

---

## [Editor Report · Acceptance letter]

24 Jan 2021

PCOMPBIOL-D-20-00703R2 

One is not enough: on the effects of reference genome for the mapping and subsequent analyses of short-reads

Dear Dr Beamud,

I am pleased to inform you that your manuscript has been formally accepted for publication in PLOS Computational Biology. Your manuscript is now with our production department and you will be notified of the publication date in due course.

With kind regards,

Alice Ellingham
